# Controlled Release Technologies for Diltiazem Hydrochloride: A Comprehensive Review of Solid Dosage Innovations

**DOI:** 10.3390/pharmaceutics17111491

**Published:** 2025-11-19

**Authors:** Estefanía Troches-Mafla, Constain H. Salamanca, Yhors Ciro

**Affiliations:** 1Grupo de Investigación en Química y Biotecnología (QUIBIO), Facultad de Ciencias Básicas, Universidad Santiago de Cali, Cali 760035, Colombia; estefania.troches00@usc.edu.co; 2Departamento de Farmacia, Facultad de Ciencias Farmacéuticas y Alimentarias, Universidad de Antioquia, Calle 70 No. 52-21, Medellín 050010, Colombia; 3Grupo de Investigación Ciencia de Materiales Avanzados, Departamento de Química, Facultad de Ciencias, Universidad Nacional de Colombia Sede Medellín, Cra. 65 #59a-110, Medellín 050034, Colombia; 4Grupo de Investigación Cecoltec, Cecoltec Services SAS, Medellín 050034, Colombia

**Keywords:** diltiazem hydrochloride, emerging technologies, methods of manufacture, modified release dosage forms

## Abstract

**Introduction**: Diltiazem hydrochloride (DH) is a calcium channel blocker used in the treatment of hypertension, angina pectoris, and arrhythmias. Its short half-life and frequent dosing requirements limit patient adherence and cause plasma concentration fluctuations. **Objective**: This review critically examines recent pharmaceutical technologies and formulation strategies for modified-release dosage forms (MRDFs) of diltiazem hydrochloride, emphasizing their impact on pharmacokinetics, clinical performance, and regulatory aspects. **Methodology**: A structured literature review (2010–2025) was conducted using databases such as PubMed, ScienceDirect, MDPI, and ACS Publications. Studies were selected based on relevance to solid oral MRDFs of DH and their associated manufacturing techniques. **Results**: Techniques including direct compression, granulation, extrusion–spheronization, spray drying, solvent evaporation, and ionotropic gelation have enabled the development of hydrophilic matrices, coated pellets, microspheres, and osmotic systems. Functional polymers such as HPMC, Eudragit^®^, and ethylcellulose play a central role in modulating release kinetics and improving bioavailability. **Conclusions:** This review not only synthesizes current formulation strategies but also explores reverse engineering of ideal release profiles and the integration of advanced modeling tools such as physiologically based pharmacokinetic (PBPK) modeling and in vitro–in vivo correlation (IVIVC). These approaches support the rational design of personalized, regulatory-compliant DH therapies.

## 1. Introduction

Diltiazem hydrochloride (DH) is a cornerstone in the pharmacological management of cardiovascular diseases, particularly hypertension, angina pectoris, and certain supraventricular arrhythmias. As a non-dihydropyridine calcium channel blocker, its primary mechanism involves the inhibition of calcium influx into cardiac and vascular smooth muscle cells during depolarization, resulting in vasodilation and reduced peripheral vascular resistance. DH also reduces heart rate and prolongs diastole, improving myocardial oxygen balance [1]. According to the Biopharmaceutical Classification System (BCS), DH is categorized as a Class I drug, characterized by high solubility and high permeability [2]. Despite its favorable solubility and permeability, immediate-release DH formulations have notable clinical limitations. These include dose-dependent adverse effects such as bradycardia and hypotension, which may compromise safety in sensitive populations [3]. Moreover, DH has a relatively short elimination half-life (approximately 3–4.5 h), requiring multiple daily doses to maintain therapeutic plasma concentrations—an issue that can negatively impact patient adherence and long-term treatment outcomes [4]. To address these challenges, the development of modified-release (MR) formulations has gained prominence. These systems aim to sustain plasma drug levels, reduce dosing frequency, and enhance both safety and therapeutic compliance [5]. Recent advances in pharmaceutical technology have introduced platforms such as pellets, microspheres, and osmotic tablets, which demonstrate superior pharmacokinetic profiles and improved tolerability compared to immediate-release counterparts [6,7]. The design and optimization of these MR formulations rely on diverse manufacturing techniques, including direct compression, wet granulation, microencapsulation, emulsification, gelification, lyophilization, and fusion granulation. Each method contributes uniquely to the modulation of drug release kinetics and the enhancement of the pharmacotechnical profile of DH [5,8,9,10].

This work synthesizes the current literature on DH-based modified-release dosage forms and their associated manufacturing strategies. By integrating biopharmaceutical, technological, and clinical perspectives, it establishes a comprehensive framework for the development of safer and more effective formulations—ultimately contributing to improved therapeutic management and patient quality of life. Beyond the descriptive analysis of formulation strategies, the scope of the review has been expanded to include the reverse engineering of ideal release profiles, aiming to align drug delivery kinetics with therapeutic needs and circadian rhythms. In addition, advanced modeling tools—such as physiologically based pharmacokinetic (PBPK) simulations and in vitro–in vivo correlation (IVIVC) frameworks—are considered for their potential to optimize formulation design, predict clinical performance, and support regulatory decision-making. This broader perspective bridges experimental development with predictive modeling, offering translational insights for the rational design of modified-release systems.

## 2. Methodology

A literature review was conducted covering the period from 2010 to 2025, using major scientific databases and repositories such as Multidisciplinary Digital Publishing Institute (MDPI), American Chemical Society (ACS) Publications, Biomed Central (BMC), PubMed, Scientific Electronic Library Online (SciELO), Sage Journals, ScienceDirect, Taylor & Francis, and Google Patents. The objective was to identify peer-reviewed studies on manufacturing methods and formulation strategies for solid modified-release forms of DH. The search algorithm and selection process are illustrated in Figure 1.

## 3. Modified Release Dosage Forms (MRDFs)

Modified release dosage forms (MRDFs) are pharmaceutical systems designed to alter the rate and/or site of release of the active substance compared to conventional formulations. Their primary objective is to maintain therapeutic plasma concentrations of diltiazem hydrochloride (DH) over extended periods, thereby reducing the frequency of administration and minimizing fluctuations in plasma levels [11,12]. MRDFs can be classified according to (i) release mechanism, (ii) release pattern, and (iii) dosage form as depicted in Figure 2.

### 3.1. Mechanism of Release

Modified-release systems can be classified according to their underlying mechanism of drug release, which includes diffusion, erosion, dissolution, swelling, osmosis, ion exchange, and bioadhesion [13]. In diffusion-controlled systems, the drug migrates through a polymeric matrix or semi-permeable membrane without inducing significant structural changes in the carrier. This migration may occur through hydrated polymer networks, insoluble frameworks, or the liquid boundary layer surrounding the device [14,15]. The process typically involves water ingress, drug dissolution, and molecular diffusion [14]. In contrast, erosion- or dissolution-based systems rely on the gradual degradation or solubilization of the matrix, whereby the drug is released as the carrier disintegrates [2,16]. Furthermore, swelling-dependent systems initiate drug release upon water absorption, which causes matrix expansion and the formation of gel layers that regulate diffusion. Polymers such as polyethylene oxide (PEO) exhibit rapid swelling behavior, significantly influencing gel formation and the duration of release [14,17,18,19,20]. Additionally, osmotic systems function through water influx across a semi-permeable membrane, generating internal pressure that forces the drug out through a calibrated orifice. These systems often employ cross-linked gelatin capsules or osmotic pumps [21,22]. On the other hand, ion exchange systems operate via reversible interactions between drug ions and resin matrices, with release rates modulated by the ionic composition of the surrounding medium [23,24]. Finally, bioadhesive systems are designed to adhere to biological surfaces, such as mucosal tissues, thereby prolonging residence time and enhancing localized drug delivery [5,25]. These mechanisms are schematically illustrated in Figure 2A, which outlines the structural and physicochemical strategies employed to modulate drug release in modified systems.

### 3.2. Pattern of Drug Release

Based on their release patterns, MRDFs can be engineered to achieve specific pharmacokinetic profiles. To begin with, sustained or prolonged release systems are designed to maintain drug concentrations within the therapeutic window over extended periods, thereby minimizing plasma fluctuations and reducing dosing frequency [2,16,19,26]. In contrast, delayed release systems postpone drug liberation until reaching a specific physiological site or time, often triggered by factors such as pH, enzymatic activity, or gastrointestinal transit. This approach enables targeted delivery and improves therapeutic efficacy [22,23]. Moreover, pulsatile release systems deliver the drug in programmed bursts, mimicking circadian rhythms or adapting to variable therapeutic demands. These systems are particularly valuable in chronotherapy or in conditions requiring intermittent dosing [27]. In addition, bimodal release platforms integrate an initial immediate-release phase for rapid onset of action with a subsequent sustained-release phase, ensuring prolonged therapeutic coverage and enhanced patient adherence [28,29]. Collectively, these release patterns—schematically illustrated in Figure 2B—reflect the strategic modulation of drug delivery kinetics aimed at optimizing pharmacological outcomes while minimizing adverse effects.

### 3.3. Pharmaceutical Dosage Form

Structurally, MRDFs fall into three primary categories—monolithic or matrix-type, reservoir-type, and multiparticulate platforms—each offering distinct strategies for controlling drug release kinetics and optimizing therapeutic outcomes. One widely employed approach is the use of monolithic or matrix-type systems, in which the drug is uniformly dispersed within a continuous polymeric material. This configuration enables release through mechanisms such as diffusion, swelling, or erosion [16,30,31]. Alternatively, reservoir-type systems encapsulate the drug within a central core surrounded by a semi-permeable or selective membrane, allowing controlled release via diffusion or osmotic pressure [32]. In addition, multiparticulate systems—including pellets [32], microspheres [25,33], nanospheres [15,34,35] and microsponges [2,10]—are composed of discrete units that can be engineered to exhibit tailored release profiles.

These units are commonly incorporated into tablets or capsules and offer enhanced flexibility in formulation design, improved gastrointestinal distribution, and reduced inter-subject variability. As illustrated in Figure 2C, the diversity of pharmaceutical dosage forms—ranging from tablets and capsules to transdermal patches—demonstrates the adaptability of these structural systems to various routes of administration and therapeutic requirements.

#### 3.3.1. Monolithic Matrix Systems

Monolithic matrix systems are pharmaceutical platforms in which the active pharmaceutical ingredient (API) is either uniformly or heterogeneously dispersed within a polymeric matrix or biocompatible excipient. Drug release from these systems is governed by the matrix structure through mechanisms such as diffusion, swelling, or erosion, with the objective of sustaining therapeutic plasma concentrations over a defined period [36,37]. The release kinetics may be modulated by diffusion through a hydrated gel layer in hydrophilic matrices, by diffusion across an insoluble framework in hydrophobic matrices, or by a combination of both in mixed systems. These platforms contribute to stabilizing plasma levels, minimizing adverse effects, and improving patient adherence (Figure 3) [33].

Recent investigations on hydrophilic matrices have compared the performance of polymers such as hydroxypropyl methylcellulose (HPMC) and polyethylene oxide (PEO) in oral delivery systems. For instance, Tsuji et al. [38] evaluated matrices composed of PEO and polyethylene glycol (PEG), reporting that PEO-based tablets released 85.7% of DH within 6 h, whereas HPMC matrices released only 63.1% over the same period. These findings underscore notable differences in hydration kinetics and gel formation between the two polymers. Furthermore, matrices formulated with HPMC K15M exhibited robust gel-forming capacity, effectively delaying DH release for up to 12 h via an anomalous diffusion mechanism involving both swelling and polymer relaxation [39].

In contrast, hydrophobic matrices offer the advantage of reducing the influence of gastrointestinal physiological variables. These systems incorporate water-insoluble polymers such as ethylcellulose and Eudragit^®^, along with lipids and waxes, to form inert structures through which the drug diffuses via channels generated during partial dissolution. The hydrophobic nature of these materials enhances resistance to pH and enzymatic fluctuations, resulting in more predictable and stable release profiles compared to hydrophilic systems [40].

Complementarily, transdermal matrix films containing DH have been developed using Eudragit RS100 in combination with HPMC K4M [33]. Eudragit RS100, being water-insoluble, enables modulation of drug release [41] and is also employed as a coating for microspheres to attenuate the initial burst release of DH. In a related strategy, Alkhamis et al. [23] formulated an insoluble matrix using polystyrene cation-exchange resins for modified DH release, eliminating the need for additional polymeric coatings. This approach exemplifies how low-permeability or insoluble matrices are essential for designing sustained-release formulations of DH.

#### 3.3.2. Multiparticulate Systems

Multiparticulate systems consist of numerous discrete units—typically microparticles ranging from 1 to 1000 μm—designed to encapsulate the API within polymeric matrices or structured cores. These units function as autonomous microreservoirs, capable of modulating drug release both temporally and spatially within the organism [42,43]. Their architecture enables customized pharmacokinetic profiles, enhanced bioavailability, and reduced local irritation, particularly in gastrointestinal and transdermal applications. Moreover, for parenteral administration, microparticles must meet strict size specifications—typically below 5 μm for intravenous use—to prevent phlebitis and vascular complications. As illustrated in Figure 4, each dosage form (e.g., capsule, gel, or injectable) contains multiple microparticles with internal reservoirs from which the API diffuses progressively over time.

This schematic representation reinforces the concept of temporal modulation of drug release and highlights the structural integrity of reservoir-based platforms. These systems are fabricated using a wide range of materials, including inorganic substrates, synthetic and natural polymers, and mineral-based excipients. Depending on the formulation strategy, they may adopt diverse structural configurations such as microspheres, micropellets, microcapsules, micro- and nanosponges, microemulsions, magnetic particles, and lipid-based vesicular systems (e.g., liposomes and niosomes) [42]. Each configuration offers distinct advantages in terms of drug loading capacity, release kinetics, and targeting potential.

##### Microsponges and Nanosponges

Microsponges and nanosponges are porous polymeric systems designed for controlled drug release. Their primary distinction lies in particle size and biological distribution: microsponges, typically ranging from 5 to 300 μm, are polymeric spheres that remain localized on tissue surfaces—such as skin, mucosa, or the gastrointestinal tract—providing extended drug release without systemic absorption [44,45]. In contrast, nanosponges, sized between 50 and 300 nm, consist of cross-linked polymeric networks (often cyclodextrin-based) capable of systemic circulation and targeted, sustained drug delivery.

In a comprehensive review, Utzeri et al. described the synthesis, structural features, and applications of cyclodextrin-based nanosponges in drug delivery and environmental systems, emphasizing how cross-linking strategies influence porosity and release behavior [46]. Complementarily, Rao et al. developed a nanosponge formulation loaded with DH for oral administration via the solvent emulsion-diffusion technique, employing ethylcellulose and β-cyclodextrin as polymeric carriers [47]. The process involved dissolving the drug and polymers in dichloromethane, followed by emulsification with an aqueous polyvinyl alcohol (PVA) phase, homogenization, filtration, and drying. SEM analysis revealed spherical, porous nanosponges with particle sizes ranging from 186 to 476 nm. In vitro release studies demonstrated that ethylcellulose enabled sustained DH release following zero-order kinetics, governed by a non-Fickian diffusion mechanism.

On the other hand, Ivanova et al. formulated topical gels containing DH-loaded microsponges for the treatment of chronic anal fissures (Figure 5) [2]. These microsponges were synthesized using Eudragit RS 100 via quasi-emulsion solvent diffusion, employing diltiazem base (poorly water-soluble) to enhance encapsulation efficiency. After encapsulation, the base was converted back to its hydrochloride salt using an HCl-generating chamber. The resulting microsponges were dispersed in two hydrogel matrices: 2% methylcellulose and 20% poloxamer 407. These formulations exhibited prolonged DH release for up to 24 h, significantly exceeding the 6–12 h release profile of conventional gels. Additionally, microsponge-based gels showed reduced permeation rates and approximately double the drug deposition in rectal mucosa compared to standard formulations.

Further insights were provided by Kolev et al., who investigated the physicochemical behavior of DH-loaded microsponges prepared with Eudragit RS 100 via quasi-emulsion solvent diffusion [10]. To elucidate drug distribution and compatibility, two reference models were employed: (i) physical drug–polymer mixtures, representing heterogeneous dispersions with surface-bound drug, and (ii) homogeneous solid dispersions obtained by molding, where the drug is molecularly dispersed within the polymer matrix. Characterization techniques included ATR-FTIR spectroscopy, differential scanning calorimetry (DSC), SEM coupled with energy-dispersive X-ray analysis (SEM-EDX), and dissolution studies. ATR-FTIR spectra indicated molecular dispersion of DH within the microsponge matrix, while DSC confirmed amorphization and compatibility. SEM-EDX revealed porous morphology and elemental distribution, suggesting partial surface localization of the drug and effective removal of residual solvents. Dissolution profiles showed that the porous architecture of microsponges enabled faster and more complete drug release compared to solid dispersions. Although this study did not culminate in a final dosage form, it established a robust analytical framework for understanding drug–matrix interactions and their impact on release kinetics.

##### Microspheres

Microspheres are solid spherical particles composed of polymeric matrices typically ranging from 1 to 1000 μm in diameter, composed of a homogeneous polymeric matrix in which the API is uniformly dispersed [11,48]. Owing to their structural simplicity and formulation versatility, they are suitable for multiple administration routes, including nasal, oral, and parenteral delivery.

In one approach, Kulkarni et al. developed nasal mucoadhesive microspheres using chitosan (CS) and DH, prepared via spray drying from aqueous solution [25]. The formulation was optimized using a central composite experimental design, adjusting polymer concentration, atomization rate, and drying temperature to maximize encapsulation efficiency (EE > 90%) and mucoadhesive performance. The resulting solid microspheres had an average diameter of ~11 μm (Figure 6A) and exhibited strong adhesion to nasal mucosa. Ex vivo permeation studies in sheep nasal tissue revealed 95.6% DH permeation after 240 min, attributed to the high degree of deacetylation and molecular weight of chitosan. Histopathological analysis confirmed preservation of epithelial architecture, with no signs of irritation or damage (Figure 6B). Accelerated stability studies over three months (ICH guidelines) showed minimal variation in particle size (10.8–12.4 μm) and EE (90.66–89.40%), with consistent mucoadhesive strength.

From an oral delivery perspective, Bolourchian et al. formulated DH-loaded microspheres using Eudragit RL and RS polymers via emulsion solvent evaporation [33]. Formulations with varying drug and polymer concentrations were characterized for yield, encapsulation efficiency, particle size, and surface morphology. Uncoated microspheres achieved EE values between 56% and 93%, with particle sizes ranging from 470 μm to over 1000 μm. Coating with Eudragit RS improved EE (82–92%) and mitigated the initial burst release. Sustained release behavior was influenced by drug-to-polymer ratio and particle size; formulations with DH–Eudragit RL ratios of 1:3 and 1:4 met USP criteria for extended-release DH products, supporting 12 and 24 h dosing intervals.

Additionally, Tareq et al. and Sahoo et al. investigated gelatin-based microsphere systems for the controlled release of DH, employing distinct formulation strategies [49,50]. Tareq et al. developed semi-interpenetrated microspheres using a natural polymer blend of sodium alginate and gelatin, stabilized with glutaraldehyde [49]. These systems demonstrated sustained DH release over 24 h, with efficiencies of 82.1% in acidic medium and 63.1% in neutral medium, following first order and Higuchi kinetics. Microspheres with an alginate–gelatin ratio of 80:20 showed optimal swelling and diffusion-controlled behavior, supporting their suitability for nasal or oral prolonged-release applications.

In contrast, Sahoo et al. formulated gelatinous microspheres via emulsification–polymerization, incorporating ethylcellulose, Eudragit RSPO, and Span 80 to enhance encapsulation and modulate release [50]. The resulting systems achieved production efficiencies of 73.33–82.53% and encapsulation efficiencies of 77.25–89.01%. In vitro studies at pH 7.4 revealed a biphasic release profile: an initial burst (~30% DH in the first hour) followed by sustained release with t_50_ values between 2.5 and 3 h. The hydrophobic polymers and glutaraldehyde crosslinking contributed to a rigid hydrogel network, reducing drug leaching, and promoting controlled release kinetics.

##### Pellets

Pellets are small, free-flowing spherical particles typically ranging from 500 to 1500 μm in diameter, produced through the controlled agglomeration of fine drug powders and excipients [51,52]. These multiparticulate units are manufactured using techniques such as extrusion–spheronization, fluid bed granulation, freeze-drying, or spray drying, yielding mechanically robust granules with uniform size and excellent flow properties. As illustrated in Figure 7, pellet-based systems exhibit notable versatility in terms of size, formulation, and packaging formats, including encapsulation in gelatin capsules, dispensing from sachets, and integration into multiple-unit pellet systems (MUPSs), which support oral sustained-release applications.

In one formulation strategy, Rashmitha et al. developed DH-loaded pellets via extrusion–spheronization, employing microcrystalline cellulose (MCC) as the structuring agent and carboxymethylcellulose (CMC) as the binder [6]. DH HCl, MCC, and CMC powders (sieved through mesh #40) were mixed for 10 min in a planetary mixer to ensure homogeneity. The dry blend was moistened with demineralized water to achieve suitable plasticity and extruded through a cylindrical roller extruder with a 1 mm nozzle, producing uniform filaments. These were processed in a spheronizer equipped with a ribbed rotor plate, where centrifugal force and controlled friction shaped them into spheres. Pellets were dried at 40 °C for 8 h, yielding particles with an average size of 1026–1061 μm, high sphericity, and favorable flow characteristics, confirmed by SEM and micromorphological analysis. The EE ranged from 89.17% to 95.27%, and in vitro release studies at pH 6.8 showed 88.1% DH release over 12 h. Stability was maintained for 90 days at 40 °C/75% RH, with no significant changes in drug content or release kinetics.

In a complementary study, Kanteti et al. formulated nine DH pellet variants via extrusion–spheronization, incorporating a hydrophobic phase of molten Gelucire 43/01 (processed at 50 °C, ground and sieved to 250 μm) combined with almond gum in varying proportions (20%, 30%, 40%), Gelucire (10%, 20%, 30%), and lactose as filler, following a 3^2^ factorial design [32]. The resulting spherical pellets ranged from 1128 to 1458 μm in size, with high EE (89–95%) and adequate mechanical strength, confirmed by SEM, DSC, and FTIR. In vitro release profiles revealed that pellets with lower Gelucire content (10%) released DH within 6 h, while intermediate formulations (20%) extended release to 8 h. The combination of 30% Gelucire and 40% almond gum achieved 88.1% release over 12 h. Statistical analysis confirmed that Gelucire primarily governed drug retention, while almond gum enhanced cohesion and reduced friability, enabling precise modulation of sustained-release profiles.

##### Nanofibers and Nanoparticles

Nanofibers and nanoparticles represent advanced platforms for controlled release, offering high surface area and tunable physicochemical properties. Nanofibers, primarily obtained via electrospinning, exhibit diameters typically ranging from 90 to 120 nm [18] or 100 to 300 nm [53], and possess a high surface-to-volume ratio that favors sustained drug release. Their structural resemblance to the extracellular matrix (ECM) enhances cellular behavior and fibroblast proliferation, making them suitable for applications such as wound dressings and tissue engineering scaffolds. In contrast, nanoparticles are colloidal solid carriers whose small size enables tissue penetration and targeted release, contributing to improved bioavailability [54].

In one formulation strategy, Etemadi et al. developed nanofibrous membranes composed of PVA, CS, and polycaprolactone (PCL), incorporating diltiazem hydrochloride (DH) at 10% *w*/*w* [53]. Electrospinning of individual polymer–drug solutions yielded flexible, porous membranes with diameters between 150 and 300 nm, reduced by the increased polarity and conductivity of the DH-containing solution. PCL, a hydrophobic polymer, enhanced mechanical strength, degradation control, and drug release modulation, while PVA and CS, both hydrophilic, facilitated faster release. FTIR confirmed chemical compatibility, and sustained DH release extended to 48 h. In vitro biocompatibility assays with L929 cells showed viability rates > 85%, reaching 99.89% in DH-loaded formulations. Fibroblasts covered nearly the entire membrane surface, suggesting favorable drug–cell interactions and enhanced proliferation. Additionally, DH incorporation improved wettability and mechanical properties, supporting the use of these nanofibers as advanced wound dressings.

In a complementary approach, Samanta et al. designed a nanocellulose-reinforced nanofiber scaffold using PVA/CS (80:20) matrices for transdermal DH delivery (Figure 8) [18]. Nanocellulose, extracted from jute waste and functionalized with ethylenediamine, was incorporated at 0.25–0.5% *w*/*w*. Electrospinning was performed at 25 kV, 0.8 mL/h, and 14 cm distance, producing smooth fibers of 150–300 nm. FTIR confirmed chemical modification, and SEM revealed uniform morphology (Figure 8A). In vitro release studies showed 67.9 ± 3.4% DH release over 72 h, compared to 92.6 ± 4.6% in scaffolds without nanocellulose, following the Korsmeyer–Peppas model. MTT assays with RAW 264.7 cells and rat skin irritation tests demonstrated >85% viability and good biocompatibility (Figure 8B). Antibacterial activity (>97% reduction in *E. coli* and *S. aureus*) was attributed to the amine groups of the nanocellulose.

Further insights were provided by Seyedian et al., who fabricated PVA nanofibrous mats (6% *w*/*v*) containing 0%, 2%, and 4% DH, electrospun at 15–17 kV and 12 cm distance and cross-linked with 10% glutaraldehyde vapor for 12 h [19]. SEM revealed fibers of ~152.7 nm and porosity of ~88.4% in the 4% DH formulation. The mats exhibited a contact angle of 29.1°, swelling of 110.4%, and in vitro release of ~94% in 120 min, fitting the Korsmeyer–Peppas model (R^2^ = 0.96, n = 0.52). In human fibroblast cultures, 4% DH mats were non-cytotoxic and increased proliferation by 263% after 5 days. In a murine wound model, lesion size was reduced to 14.7% after 14 days, with improved histopathological parameters and reductions in malondialdehyde (−63%) and nitrite (−59%) levels.

Representative examples of nanoparticles in DH formulations include CS conjugated with L-leucine for pulmonary inhalation, synthesized via water-in-oil emulsification and glutaraldehyde crosslinking [34]. These nanoparticles exhibited a leucine substitution degree of 0.89, creating an amphiphilic environment that enhanced swelling and prolonged DH release at pH 7.3. Similarly, Khairnar et al. developed CS nanoparticles loaded with repaglinide and DH via ion gelation with tripolyphosphate, producing particles of 200–300 nm with sustained release of ~19 h for repaglinide and ~20 h for DH, without burst effect [54]. Lastly, modified silica nanoporters (Silica-03) functionalized with silane KH-570 demonstrated superior DH adsorption and extended release compared to unmodified silica (Silica-01), maintaining crystal stability per DSC and FTIR, due to enhanced hydrophobic affinity from the organofunctional coating.

##### Niosomes

Niosomes are multiparticulate vesicular systems formed by the self-assembly of non-ionic surfactants—such as Spans, Brijs, and polysorbates—in combination with cholesterol as a stabilizing excipient, which can encapsulate hydrophilic drugs within their internal aqueous core and lipophilic drugs within the lipid bilayer, offering high physical and chemical stability during preparation and storage, as well as versatility for the controlled release of molecules with diverse polarity profiles [55]. As illustrated in Figure 9, freeze-dried niosomal powder can be reconstituted in aqueous dispersion media to form bilayer vesicles capable of dual drug loading and targeted release. Importantly, such reconstitution requires that the vesicles have been previously formed and stabilized—commonly via spray drying or lyophilization—since non-ionic surfactants alone do not spontaneously self-assemble into uniform bilayer structures upon simple aqueous dispersion.

Niosomes are typically prepared by methods such as thin film hydration, ether injection, reverse-phase evaporation, and microfluidic techniques, each offering distinct control over vesicle size, lamellarity, and encapsulation efficiency. Thin film hydration is widely employed for its simplicity and scalability, while ether injection and reverse-phase evaporation allow finer control of bilayer formation and drug entrapment. Microfluidic approaches, although more recent, enable precise modulation of particle size and polydispersity through controlled flow dynamics. Regardless of the method, the resulting niosomes generally exhibit diameters between 10 nm and 1 μm, low polydispersity, and high biocompatibility. Their release kinetics can be modulated by adjusting the surfactant-to-cholesterol ratio [56].

In a complementary design, Ammar et al. developed DH-loaded niosomes for nasal administration using the thin film hydration technique [57]. Non-ionic surfactants (Span 60 or Brij-52) and cholesterol were dissolved in a chloroform/methanol mixture (2:1 *v*/*v*) to form a thin lipid film. Following solvent evaporation via rotary evaporation, the film was rehydrated with phosphate buffer (pH 7.4) at 60 °C for 30 min and matured overnight at 4 °C. The resulting suspension contained spherical unilamellar vesicles with mean sizes ranging from 0.82 to 1.59 μm. In vivo pharmacokinetic studies in male Wistar rats revealed a significant increase in mean residence time, elimination half-life, and area under the curve (AUC), along with a decrease in the elimination constant, indicating prolonged therapeutic action and enhanced bioavailability of DH compared to control solutions.

In a complementary approach, Akbari et al. formulated a topical DH-loaded niosomal gel using the ultrasonic method, optimizing the cholesterol-to-surfactant ratio to achieve vesicles with controlled particle size, low polydispersity index, and high encapsulation efficiency (~94%) [35]. In vitro release studies demonstrated sustained DH release over 24 h. In vivo pharmacological evaluation in an animal wound model showed that the niosomal gel achieved wound closure efficacy comparable or superior to a commercial reference, with enhanced tissue regeneration and no signs of skin irritation. Histopathological analysis and skin sensitivity tests confirmed the therapeutic potential of this nanoparticulate platform for dermal applications.

#### 3.3.3. Reservoir-Type

Reservoir-type formulations are strategic platforms for modulating the rate and timing of drug release, thereby improving bioavailability and patient adherence. These systems are defined by the presence of a membrane or barrier that regulates drug diffusion into the surrounding medium.

A representative example for DH was reported by Ishrath et al., who developed a PulsinCap^®^-type system consisting of hard gelatin capsules cross-linked with formaldehyde (12 h vapor exposure followed by drying) [27]. The capsules contained DH granules obtained via wet granulation and a hydropolymer plug (HPMC and ethylcellulose) that controlled the lag time. Optimization of plug composition revealed that 90 mg produced a 3–4 h delay, while 100 mg extended it to 7–11 h, depending on polymer ratios. The PF5 formulation (HPMC 50 mg/ethylcellulose 50 mg) was selected for its minimal release during latency and rapid post-delay release. Accelerated stability studies confirmed consistent performance, demonstrating that capsule cross-linking and plug engineering enable precise temporal control—ideal for chronotherapy and targeted administration. Additionally, another reservoir strategy is the “tablets-in-a-capsule” technique, where the capsule shell acts as a barrier layer that dissolves or degrades, releasing the drug from internal tablets arranged as a multiparticulate core [9].

A specialized subcategory of reservoir systems is the osmotic pump, illustrated in Figure 10. These systems rely on water permeation through a semipermeable membrane, which hydrates the drug-loaded core and generates internal pressure that expels the drug through a precisely sized orifice, achieving zero-order kinetics and minimizing variability due to pH or intestinal motility. In the elementary osmotic pump (EOP), drug granules are compressed with hydrophilic excipients and coated with cellulose acetate, followed by orifice drilling. Upon hydration, the core expands and steadily releases the drug. In controlled porosity osmotic pumps (CPOP), the coating includes soluble porogens (e.g., PEGs) that dissolve in situ to form microchannels, eliminating the need for mechanical drilling. In modulated solubility osmotic pumps (SMOP), salts such as NaCl are added to reduce drug solubility, maintaining a constant osmotic gradient, and extending release up to 16–18 h [21].

In one formulation strategy, Joshi et al. applied a Quality by Design (QbD) approach to develop DH EOP tablets via wet granulation, using HPMC and cellulose acetate coatings [58]. Optimization of polymer ratios and coating weight yielded zero-order release for 14–16 h, with robust mechanical properties and strong in vivo correlation. In contrast, Soliman and Ibrahim formulated CPOP tablets by mixing DH with mannitol (1:1.5) as an osmotic agent, followed by coating with Eudragit RS-100 and PEG-400 [59]. Upon PEG dissolution, pores formed that enabled ~90% release in 8 h, independent of pH or agitation. In a complementary study, they investigated two key variables: (i) coating thickness, which modulates water ingress and osmotic pressure, and (ii) PEG molecular weight, which affects porosity and release rate. Thicker coatings delayed release, while low molecular weight PEGs produced more porous membranes and faster release. The optimal balance achieved sustained zero-order kinetics, demonstrating the importance of membrane engineering in reservoir systems.

Further refinement was achieved by Monton and Kulvanich, who designed a multi-chamber push-pull osmotic pump (PPOP) for oral DH delivery [22]. The system used formaldehyde-crosslinked gelatin capsules containing a drag layer (10 mg DH, 130 mg PEO Mw 200K, 40 mg lactose) and a push layer (55 mg PEO Mw 5000K, 34.5 mg NaCl, 0.5 mg pigment), coated with HPMC/PEG 4000 and eight layers of cellulose acetate (6% *w*/*w*), ending with a 0.6 mm outlet orifice. Upon hydration, water penetrates both chambers, dissolving the drug and expanding the polymer, generating pressure that ejects the DH suspension with zero-order kinetics, independent of pH, stirring, or solubility. The formulation remained stable for 12 months, with consistent Higuchi-model release and latency times, supporting its industrial scalability.

Table 1 presents a comparison of the advantages and disadvantages of MDRFs compared to conventional dosage forms.

In summary, the formulation of DH MDRFs has enabled precise modulation of pharmacokinetic behavior, improved therapeutic adherence, and broadened the range of administration routes. Through platforms such as matrix tablets, pellets, microspheres, nanofibers, and osmotic systems, researchers have addressed the limitations associated with DH’s short half-life, variable bioavailability, and chronopharmacological requirements. These advances reflect the strategic integration of polymer selection, structural design, and release mechanisms to optimize clinical performance. Building on these formulation principles, the following section outlines the general manufacturing methods employed in MRDFs development, emphasizing key unit operations, process variables, and scale-up considerations.

## 4. General Methods of Manufacture of MRDFs

The manufacture of MRDFs requires specialized technologies and process controls to ensure the quality, reproducibility, and functional performance of the final product. Figure 11 illustrates the principal manufacturing methods employed in the pharmaceutical industry for solid MRDFs platforms, highlighting their relevance across formulation types and dosage formats.

### 4.1. Optimal Pharmacokinetics and Integration with Drug Delivery Technology

An optimal controlled release (CR) formulation of DH should maintain plasma concentrations within the therapeutic window—typically 50 to 200 ng/mL—over an extended period, ideally 24 h. This target aims to minimize peak–trough fluctuations, reduce dose-dependent adverse effects such as bradycardia and hypotension, and improve adherence in chronic cardiovascular therapy. Achieving zero-order release kinetics is considered ideal, as it ensures a constant drug input rate independent of physiological variables. Recent studies have demonstrated the feasibility of approximating this ideal profile using scalable technologies and validated modeling approaches. Joshi et al. (2022) [58] developed an osmotic DH formulation using a Quality by Design (QbD) strategy, achieving near zero-order kinetics and robust performance across physiological conditions. Arafat et al. (2021) [72] evaluated a polymer matrix-based DH formulation against a commercial reference (Tildiem^®^), showing sustained release up to 10 h and comparable pharmacokinetic parameters, reinforcing the clinical viability of matrix systems. Complementarily, Taha and Emara (2022) [73] applied convolution and deconvolution modeling to predict plasma profiles of extended-release DH tablets, achieving strong in vitro–in vivo correlation (IVIVC) and minimal prediction error for Cmax and AUC.

These findings support the rational design of CR formulations that approximate ideal pharmacokinetic behavior, combining therapeutic consistency with industrial scalability. To translate these pharmacokinetic objectives into functional dosage forms, the selection and optimization of manufacturing technologies becomes critical. The ability to approximate zero-order kinetics or sustain therapeutic plasma levels over extended periods depends not only on the choice of excipients and release mechanisms, but also on the precision and reproducibility of the underlying processes. The following subsections outline the principal manufacturing methods employed in the development of MRDFs, highlighting their relevance in aligning formulation performance with pharmacokinetic targets.

### 4.2. Wet and Dry Granulation

Wet granulation (Figure 11A) remains widely used in DH MRDF development for enhancing compressibility and flow. This method involves the agglomeration of powder particles using a liquid binder to form granules with enhanced compressibility, flowability, and uniformity. Its versatility has enabled the incorporation of natural polymers—such as karaya gum and carob gum—as matrix formers, with polyvinylpyrrolidone K-30 (PVP K-30) serving as a binder to modulate drug release kinetics [74]. In parallel, pectin (PEC) has been employed as a matrix material in wet-granulated tablets to achieve controlled release profiles, demonstrating the adaptability of this technique to diverse excipient functionalities [75].

By contrast, dry granulation (Figure 11B) is preferred when the API or excipients exhibit sensitivity to moisture, as it eliminates the need for liquid binders. Techniques such as roller compaction have been successfully applied to produce sustained-release tablets containing hydrophilic and hydrophobic polymers—including HPMC, hydroxypropylcellulose (HPC), and methacrylic acid copolymers—ensuring mechanical integrity and controlled release without compromising chemical stability [76]. Moreover, dry processing has proven particularly advantageous in MRDFs systems based on cation exchange resins, where moisture-free conditions help preserve drug stability and optimize powder flow and compressibility [23].

### 4.3. Direct Compression

Direct compression (Figure 11C) is one of the most efficient and scalable techniques for tablet production, as it eliminates intermediate steps such as granulation and drying, thereby reducing processing time and operational complexity [77]. This approach requires excipients with excellent flowability and compressibility, including MCC, dicalcium phosphate dihydrate, and spray-dried lactose, which facilitate uniform die filling and robust tablet formation [78].

In the context of MRDFs, direct compression has been successfully employed using HPMC matrices of varying viscosities. These polymers serve dual functions: they act as compression aids during tableting and as release modulators within the gastrointestinal tract, enabling sustained drug delivery profiles [79]. Furthermore, the absence of moisture and heat exposure during processing helps preserve drug integrity, particularly for thermolabile or hydrolytically sensitive compounds. By minimizing process variables, direct compression also enhances batch-to-batch reproducibility and formulation robustness, making it a valuable strategy for MRDFs development under both pilot and industrial conditions.

### 4.4. Ionic Gelification

Ionic gelification (Figure 11D) refers to the capacity of specific polymers to form a semi-solid, three-dimensional network that entraps the drug, thereby creating a matrix system capable of modulating its release profile [80]. Depending on the triggering mechanism, gelification processes can be classified as ionotropic, thermal, covalent, or pH-sensitive, each offering distinct structural and kinetic characteristics.

For diltiazem hydrochloride formulations, MRDFs, ionotropic gelification has been extensively investigated. This technique employs anionic polymers such as sodium alginate, which undergo gelation upon contact with multivalent cations (e.g., Ca^2+^, Zn^2+^, Fe^3+^). Typically, DH is dispersed in an alginate solution and dropped into a calcium chloride bath, where ionic crosslinking transforms the droplets into spherical microspheres. The resulting gelled particles exhibit mechanical integrity and sustained release behavior, making them suitable for MRDFs applications under gastrointestinal or mucosal conditions [80,81]

### 4.5. Hot-Melt Extrusion (HME)

HME (Figure 11E) enables continuous mixing and shaping of drug–polymer blends, allowing controlled release and integration with 3D printing for personalized dosage design.

A notable advancement in this domain is the integration of HME with 3D Fused Filament Deposition (FDM) printing, where HME enables the fabrication of drug-loaded filaments suitable for additive manufacturing. These filaments, incorporating DH and polymers such as HPMC, have demonstrated thermal stability and programmable release profiles governed by print geometry, infill density, and surface area-to-volume ratios [82]. As such, HME serves not only as a conventional MRDFs production method but also as a foundational technology for personalized pharmaceutical manufacturing via 3D printing, offering precise control overdose, release kinetics, and spatial distribution of the API [83,84].

### 4.6. Spray Drying

Spray drying (Figure 11F) is a widely employed technique for converting liquid formulations—such as solutions, suspensions, or emulsions—into dry solid particles via atomization in a heated air stream. This method is particularly valuable for microencapsulation and the production of spherical microparticles with uniform size distribution, enhanced flow properties, and compatibility with thermolabile drugs, owing to rapid solvent evaporation and minimal thermal exposure [78]. Key process parameters—including inlet and outlet temperatures, feed rate, atomization pressure, and solvent composition—enable precise control over particle morphology, encapsulation efficiency, and release behavior.

For DH-containing MRDFs systems, spray drying has been applied using polymers such as Eudragit L100, HPMC K4M, guar gum, and xanthan gum. A recent study optimized a formulation based on Eudragit L100 (drug-to-polymer ratio 1:2 in isopropanol), achieving controlled release over 12 h, content uniformity of approximately 70%, and zero-order kinetics [85]. These findings underscore the versatility of spray drying in tailoring release profiles and enhancing the physicochemical stability of diltiazem hydrochloride formulations.

### 4.7. Extrusion-Spheronization

Extrusion–spheronization (Figure 11G) is a two-step manufacturing process designed to produce spherical pellets with consistent size, shape, and mechanical integrity. Initially, a wet mass containing DH and selected excipients—such as polymers, waxes, or natural gums—is extruded through a perforated die to generate cylindrical extrudates. These are subsequently processed in a spheronizer, where rotational motion and frictional forces reshape them into uniform spheres [86]. Key process variables—including rotation speed, frictional energy, and moisture content—directly influence pellet morphology, surface characteristics, and drug release kinetics.

This technique has been successfully applied to develop sustained-release DH pellets using excipients such as gelucire 43/01 and almond gum. Resulting formulations demonstrated high production yields (86–92%) and prolonged drug release, confirming the suitability of extrusion–spheronization for MRDFs systems requiring precise control over particle architecture and release dynamics [32].

### 4.8. Pharmaceutical Coating

Pharmaceutical coating (Figure 11H) is a versatile and widely adopted technique for modifying both the site of drug release and its kinetic profile, particularly in solid dosage forms such as tablets, pellets, and multiparticulate systems. Among the most established approaches is fluid-bed coating, wherein particles are suspended in an upward air stream while a polymeric solution is sprayed. This method commonly employs polymers such as ethylcellulose for sustained release, hydroxypropyl methylcellulose phthalate (HPMCP) for enteric protection, and methacrylic acid copolymers like Eudragit L100/S100 for pH-dependent release.

An alternative strategy is perforated drum coating, in which the cores rotate while the coating solution or suspension is applied. This configuration allows precise control of temperature and humidity, making it particularly suitable for formulations involving organic solvents. Additionally, the layering technique on inert nuclei—such as cellulose microspheres or sugar spheres—enables the sequential deposition of drug and excipients, allowing fine-tuned control over content uniformity and release modulation [87,88].

For diltiazem hydrochloride formulations, various coating strategies have been explored. One example involves aqueous HPMC coatings derived from raw cotton, successfully applied to compressed tablets. In this study, different viscosity grades of HPMC (18–52 cps) and coating levels (2–8%) were evaluated, demonstrating that higher polymer viscosity and thicker films markedly delayed drug release, in contrast to uncoated tablets which released ~96% of the drug within 1 h [89]. Alternatively, fluid-bed coating of DH pellets with Eudragit FS 30D at 35% *w*/*w* has been implemented to achieve a 12 h pulsatile release profile under variable pH conditions [90]. These examples underscore the potential of coating technologies to design programmable and environmentally responsive MRDFs systems.

### 4.9. Evaporation of Solvent

Solvent evaporation (Figure 11I) is a widely employed technique in pharmaceutical microencapsulation, particularly for the development of MRDFs [91,92]. Grounded in mass transfer principles, this method utilizes water-insoluble polymers as encapsulating matrices to form microparticles or microspheres with controlled release properties. The process typically involves the creation of a two-phase system, wherein the drug and polymer are dissolved in a volatile organic solvent to form a homogeneous solution. This solution is then emulsified into an aqueous phase containing surfactants under high-shear homogenization.

The core mechanism relies on the diffusion of the organic solvent from the emulsion droplets into the continuous aqueous phase. As the solvent migrates, the polymer solubility decreases, triggering its controlled precipitation around the drug. This results in the formation of encapsulated particles with tailored morphology and release kinetics [93].

For DH MRDFs systems, solvent evaporation has been successfully applied using polymers such as poly(lactic-co-glycolic acid) (PLGA), ethylcellulose, and Eudragit derivatives. These encapsulating agents enable the design of sustained and programmable release profiles, while preserving drug stability and enhancing formulation versatility [94].

### 4.10. Combination of Manufacturing Methods

In contemporary pharmaceutical development, most diltiazem hydrochloride formulations are produced through combinations of manufacturing techniques, allowing for the optimization of drug performance and formulation robustness. This integrated approach leverages the strengths of individual methods while mitigating their limitations, enabling the design of complex systems with tailored release profiles.

Representative examples include matrix systems incorporating polymers such as poloxamer-188 in combination with HPMC, which modulate hydration dynamics and diffusion rates [72]. Other strategies involve multi-layer coated pellets produced via extrusion–spheronization, using MCC as a spheroizing agent, followed by sequential coatings—HPMC for core protection and ethylcellulose for sustained release control. Additionally, PEO of varying molecular weights has been employed to fine-tune matrix behavior and drug release kinetics [16].

Among the most impactful innovations are osmotic technologies, which have significantly advanced the design of DH MRDF. These include osmotic tablets and pumps capable of delivering drugs with zero-order kinetics, independent of pH and gastrointestinal motility [21]. Recent developments have integrated 3D printing—particularly semi-solid extrusion—to fabricate tablets with programmable osmotic cores and semipermeable shells [95]. Furthermore, push–pull systems combining enteric coatings and ion-exchange resin complexes offer dual-release mechanisms, enhancing bioavailability and therapeutic precision. These hybrid platforms complement conventional osmotic systems and represent a cornerstone in the evolution of modified-release technologies for DH [96].

### 4.11. Emerging Technologies: 3D Printing

Three-dimensional (3D) printing, also referred to as additive manufacturing, has emerged as a disruptive technology in pharmaceutical development, enabling the fabrication of personalized solid dosage forms through computer-aided design and layer-by-layer deposition. Among the pioneering platforms, ZipDose^®^ technology stands out as the first formulation system to employ 3D printing for drug manufacturing, utilizing aqueous fluid binding of powdered layers without reliance on compression forces, punches, or dies [97,98]. This approach enables the production of highly porous structures with rapid disintegration and precise dose control.

Recent studies demonstrate the potential of 3D printing to fabricate multi-drug dosage forms with programmable release [99]. In diltiazem hydrochloride formulations, one study reported the integration of 3D printing with hot-melt extrusion (HME) to fabricate dual filaments: one composed of PVA, DH, and osmogens (NaCl and/or mannitol) for the osmotic core; and another of cellulose acetate for the semipermeable shell [100]. Using a dual-extruder fused deposition modeling (FDM) printer, the authors constructed core–shell units with engineered geometries, including micro-orifices to facilitate water ingress and lateral weakening cavities designed to rupture under osmotic pressure.

The release onset was programmable at 0, 120, or 360 min, depending on the structural configuration of the shell, followed by sustained drug release. Physicochemical characterization of the filaments and printed constructs included TGA, DSC, mechanical testing, and X-ray diffraction (XRD), confirming DH amorphization post-HME/FDM and retention of NaCl crystallinity. In situ monitoring via micro-computed tomography (micro-CT) revealed volumetric expansion of the osmotic core during hydration, leading to targeted rupture of the shell and transition from latency to controlled release.

This highlights the potential of HME-FDM integration for personalized MRDFs fabrication, where geometric modulation of the dosage form is as critical as its composition in determining the release profile. The integration of HME and FDM printing thus represents a promising avenue for the development of programmable, patient-specific therapies.

### 4.12. Current Commercial Formulations of DH

Several extended-release formulations of DH are currently available on the pharmaceutical market, including Cardizem CD^®^, Cardizem LA^®^, Cartia XT^®^, Tiazac^®^, and Dilacor XR^®^ in the United States, as well as Tiazac XC^®^ in Canada [101,102]. Although all products are presented as modified-release capsules or tablets, they differ in the technological platforms employed to regulate drug release. For instance, Cardizem CD^®^ utilizes a multiparticulate system composed of microgranules coated with semipermeable polymers, whereas Cartia XT^®^ and Tiazac^®^ rely on hydrophilic matrix systems that swell upon hydration to form a gel barrier, thereby modulating drug diffusion [103].

These technological distinctions result in varied pharmacokinetic profiles. Formulations such as Cardizem LA^®^ and Tiazac^®^ provide sustained release over 24 h, enabling once-daily administration, while others—including Cartia XT^®^—exhibit 12 h release, requiring twice-daily dosing. Despite these differences, all formulations demonstrate clinical bioequivalence in the management of hypertension and angina pectoris, with 24 h systems offering improved therapeutic adherence due to simplified dosing regimens [101,103].

From a pharmacotechnical perspective, MRDFs of DH offer multiple advantages that contribute to therapeutic optimization. The reduction in dosing frequency—from three or four times daily to one or two—simplifies administration and enhances patient compliance. These systems also minimize fluctuations in plasma concentrations, thereby reducing the incidence of adverse effects associated with peak levels. The incorporation of protective matrices or coatings confers stability against acid-mediated degradation, improving drug performance in gastric environments. Furthermore, MRDFs platforms enable chronomodulated release, which is particularly relevant in the treatment of hypertension—a condition characterized by circadian variation and early morning blood pressure surges. Finally, gradual or site-specific release along the gastrointestinal tract contributes to the attenuation of first-pass metabolism, enhancing systemic bioavailability and ensuring more consistent therapeutic outcomes.

To consolidate the diverse formulation strategies and technological platforms described throughout this section, Table 1 presents a structured overview of the principal investigations focused on MRDFs of DH. The table highlights key aspects of each study, including the manufacturing method employed, the therapeutic objective pursued—such as sustained, pulsatile, or chronomodulated release—and the compositional features that define each system. This synthesis facilitates comparative analysis across approaches, offering insight into how polymer selection, process variables, and structural design converge to optimize drug performance and clinical applicability.

### 4.13. Industrial Viability and Comparative Implementation Analysis

Although MRDFs offer significant therapeutic advantages, their industrial viability depends on factors beyond pharmacokinetic performance, such as scalability, cost-efficiency, compatibility with existing manufacturing infrastructure, and regulatory compliance [104,105]. Osmotic pump technologies are widely recognized for delivering drugs at a near zero-order rate, independent of physiological variables like pH, motility, and food intake. Their modular design and membrane-controlled release make them highly reproducible and scalable [21]. More recent innovations, such as magnetic nanoparticles, have enhanced their performance, supporting long-term use and reducing dosing frequency, which could be a promising strategy for this active pharmaceutical ingredient.

Polymer matrix systems—particularly hydrophilic and hydrophobic matrices—represent an attractive alternative due to their simplicity, cost-effectiveness, and compatibility with conventional tableting equipment. These systems allow flexible modulation of drug release and are suitable for hot-melt extrusion, solvent evaporation, and compression molding. Advances in polymer science and 3D printing further reinforce their scalability and personalization potential [106]. Regulatory acceptance of matrix systems is well established, especially when supported by dissolution testing and bioavailability studies.

Additionally, Level A IVIVC modeling has enabled the linkage of dissolution profiles with clinical pharmacokinetics, making virtual bioequivalence studies and regulatory approvals possible without extensive in vivo trials. Although it requires robust data, its integration can accelerate development and facilitate post-approval adjustments. Mechanistic models such as PBPK are also gaining acceptance by agencies like the FDA and EMA for complex formulations [105].

To provide a comprehensive perspective, this analysis also considers the full range of manufacturing methods described in Section 4, organized from conventional to emerging technologies. These include wet and dry granulation [107], direct compression [108], ionic gelification [109], hot-melt extrusion [110], spray drying [111], extrusion-spheronization [112], pharmaceutical coating, solvent evaporation, and hybrid or additive manufacturing approaches such as 3D printing. Each method presents distinct advantages and limitations in terms of industrial scalability, equipment compatibility, cost, and regulatory alignment. Table 2 summarizes the industrial possibilities of each platform to support formulation selection and technology transfer decisions.

### 4.14. Market Traction and Bibliometric Visualization

To complement the technological and pharmacokinetic analysis of controlled release strategies for DH, this section addresses the financial and scientific traction of the field. From a market perspective, DH remains a relevant molecule in the cardiovascular segment, with sustained-release formulations representing a niche but stable share of the total addressable market (TAM). While generic immediate-release products dominate in terms of volume, controlled-release formats—particularly once-daily tablets—offer differentiated value in terms of adherence and therapeutic consistency, justifying their continued development in select patient populations and regulatory environments [113,114]. To visualize the scientific traction of controlled release technologies applied to DH, a bibliometric analysis was conducted using VOSviewer software version 1.6.20. The co-occurrence map (Figure 12) was generated from Scopus-indexed publications between 2000 and 2025, filtered by keywords such as Diltiazem hydrochloride, Emerging technologies, Methods of manufacture, and Modified release dosage forms.

From this bibliometric visualization, it becomes evident that while Diltiazem hydrochloride remains a central node in the co-occurrence network, the color gradient in visualization (A)—coded by average publication year—shows that most relevant studies were published prior to 2020. Dominant purple tones around keywords such as controlled release, microspheres, pharmacokinetics, and drug delivery suggest that research activity on MRDFs of DH has declined in recent years. This trend may reflect technological maturity, a shift toward newer therapeutic agents, or reduced editorial prioritization in high-impact journals.

In contrast, visualization (B)—coded by average citation count—indicates that the most cited studies are also associated with established technologies, reinforcing their historical relevance but also highlighting a lack of thematic renewal. This temporal and thematic gap underscores the need for a critical review like the present one, which consolidates accumulated knowledge and connects it with emerging platforms and industrial feasibility criteria.

Based on this analysis, Table 3 provides a structured synthesis of key investigations on modified-release dosage forms of DH, organized by system type, manufacturing method, therapeutic application, and core composition.

## 5. Conclusions

MRDFs of DH offer a practical solution to overcome the drug’s short half-life and the need for frequent dosing. The reviewed studies show that various manufacturing techniques—such as direct compression, granulation, extrusion–spheronization, spray drying, solvent evaporation, and gelation—have enabled the development of platforms like matrices, pellets, microspheres, osmotic systems, and transdermal devices. These approaches have been effective in controlling drug release, improving bioavailability, and enhancing patient adherence.

From a formulation point of view, the use of functional polymers—including HPMC, Eudragit^®^, ethylcellulose, and chitosan—has played a key role in adjusting release profiles under different physiological conditions. In addition, combining manufacturing methods has proven useful for creating multiparticulate systems and matrix-based formulations with better control over release kinetics and mechanical properties.

On the clinical side, MRDFs of DH help maintain stable plasma levels, reduce side effects related to peak concentrations, and support chronotherapy in conditions like hypertension. These benefits contribute to better treatment outcomes and improved patient compliance.

Looking ahead, technologies such as hot-melt extrusion combined with 3D printing, and systems based on nanomaterials, show promise for developing personalized and programmable MRDFs. These innovations allow precise control over dosage form geometry and release behavior, aligning with current trends in precision medicine. Moreover, the rational design of these systems increasingly involves the reverse engineering of ideal release profiles and the integration of advanced modeling tools—such as physiologically based pharmacokinetic (PBPK) simulations and in vitro–in vivo correlation (IVIVC)—to predict clinical performance and support regulatory decision-making. Further research and industrial application will be important to bring these advanced systems into routine clinical use.

## Figures and Tables

**Figure 1 pharmaceutics-17-01491-f001:**
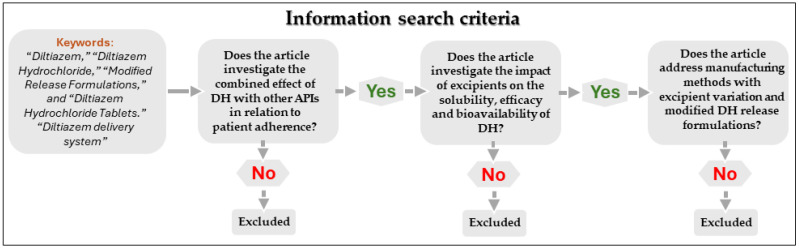
Information search algorithm.

**Figure 2 pharmaceutics-17-01491-f002:**
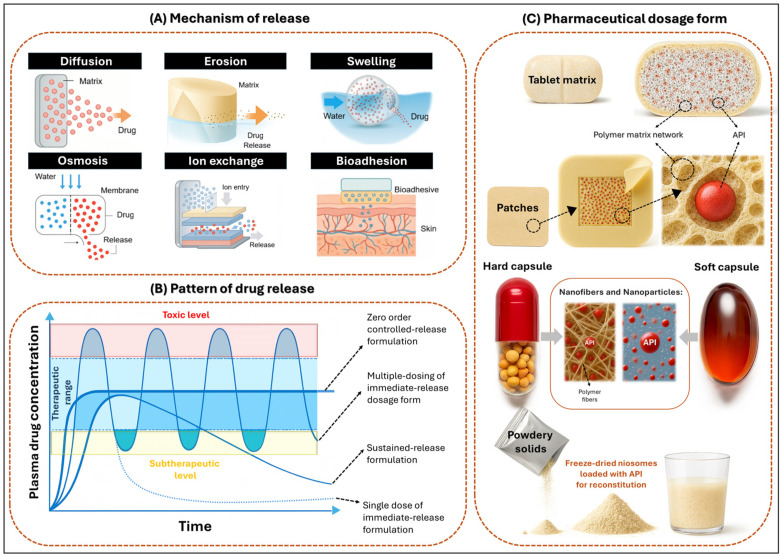
Conceptual framework for MRDFs. (**A**) Principal mechanisms of drug release, including diffusion, erosion, swelling, osmosis, ion exchange, and bioadhesion. (**B**) Representative plasma concentration–time profiles illustrating immediate, sustained, delayed, and pulsatile release patterns in relation to therapeutic windows. (**C**) Pharmaceutical dosage forms commonly employed in MRDFs, such as matrix tablets, transdermal patches, capsules, powders, nanofibers, and nanoparticles.

**Figure 3 pharmaceutics-17-01491-f003:**
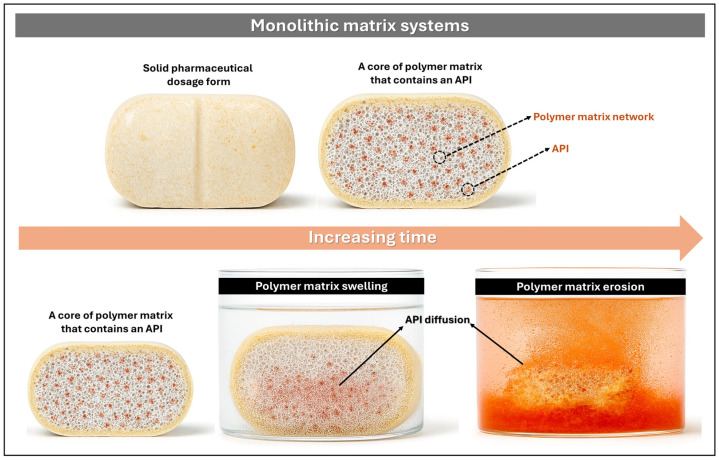
Schematic representation of drug release from monolithic matrix systems, illustrating the sequential processes of swelling, erosion, and diffusion of the API into the surrounding medium.

**Figure 4 pharmaceutics-17-01491-f004:**
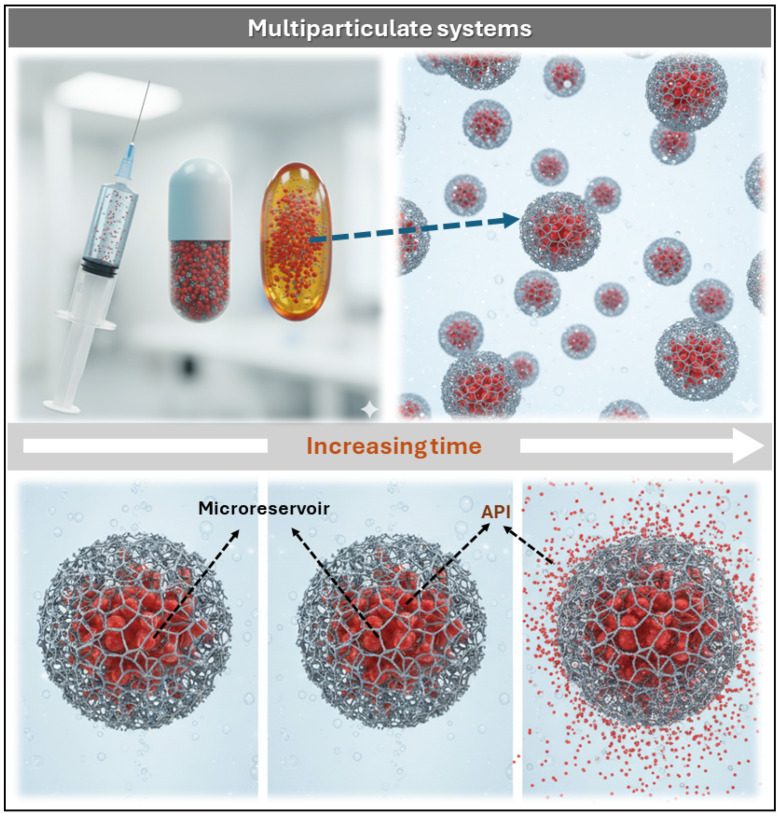
Schematic representation of multiparticulate systems: dosage forms containing microparticles with polymeric microreservoirs that release the API progressively over time.

**Figure 5 pharmaceutics-17-01491-f005:**
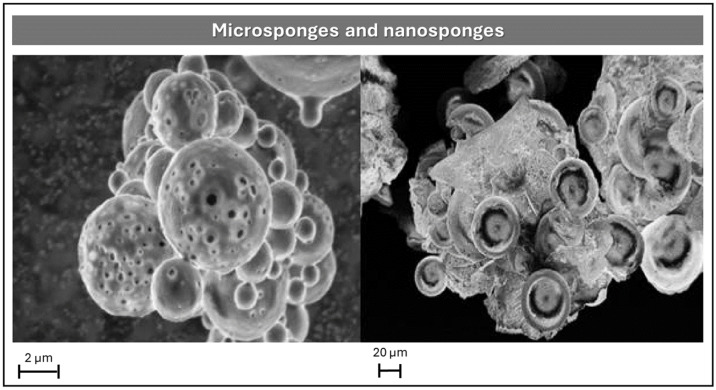
SEM micrographs comparing microsponges and nanosponges: porous polymeric spheres (**left**, 2 μm scale) designed for localized drug delivery, and cross-linked nanoscale structures (**right**, 20 μm scale) intended for systemic administration. Figures Reprinted from Nadezhda Antonova Ivanova, Adriana Trapani, Cinzia Di Franco, Delia Mandracchia, Giuseppe Trapani, Carlo Franchini, Filomena Corbo, Giuseppe Tripodo, Iliyan Nikolov Kolev, Georgi Stoyanov Stoyanov, Kameliya Zhechkova Bratoeva, In vitro and ex vivo studies on diltiazem hydrochloride-loaded microsponges in rectal gels for chronic anal fissures treatment, Pages No. 62, Copyright (2025), with permission from Elsevier [2].

**Figure 6 pharmaceutics-17-01491-f006:**
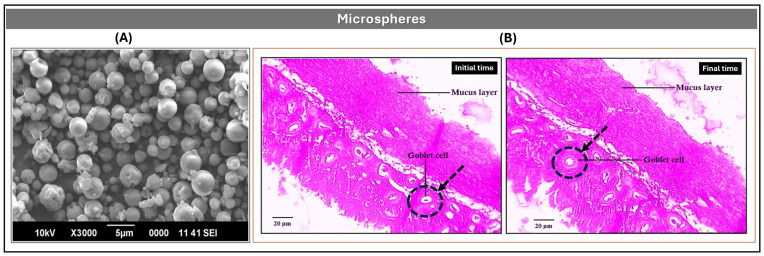
(**A**) Optimized DH-loaded microsphere formulation (D7); (**B**) Optical photomicrographs of untreated sheep nasal mucosa and mucosa treated with 100 mL of 1% *v*/*v* DH microsphere suspension for 7 h. Kulkarni, Deepak B. Bari, Sanjay J. Surana, Chandrakantsing V. Pardeshi, In vitro, ex vivo and in vivo performance of chitosan-based spray-dried nasal mucoadhesive microspheres of diltiazem hydrochloride, Pages No. 113 and 115, Copyright (2016), with permission from Elsevier [25].

**Figure 7 pharmaceutics-17-01491-f007:**
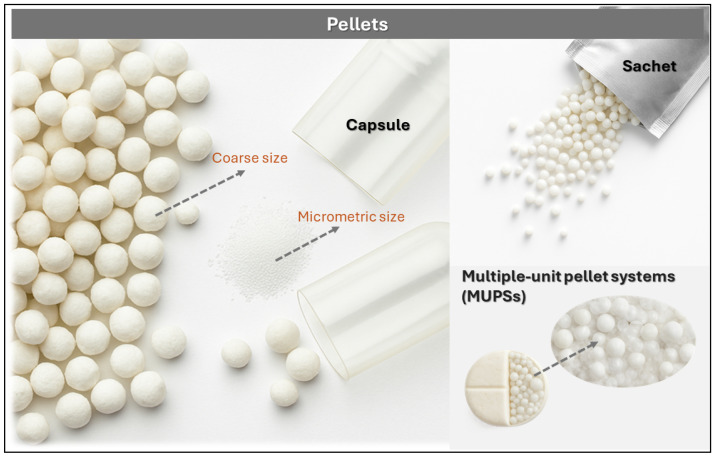
Visual representation of pellet-based drug delivery systems: coarse and micrometric pellets encapsulated in gelatin capsules, dispensed from sachets, and integrated into multiple-unit pellet systems (MUPSs). These formats illustrate the structural diversity and packaging flexibility of sustained-release pellet formulations.

**Figure 8 pharmaceutics-17-01491-f008:**
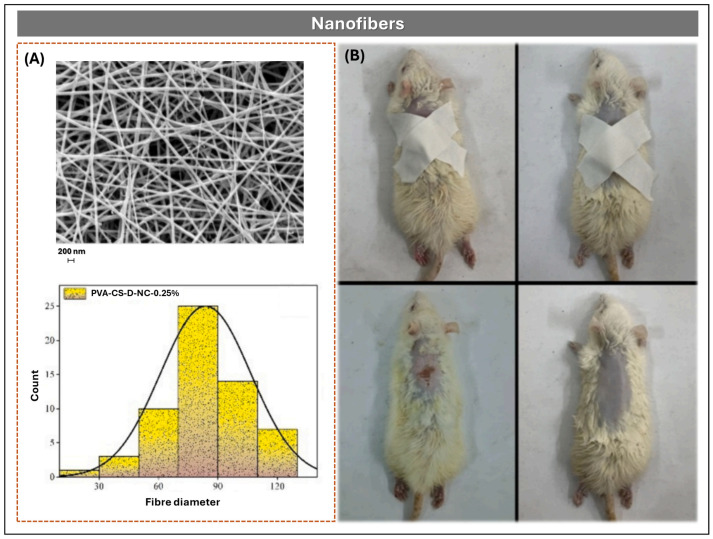
(**A**) SEM micrograph and diameter distribution histogram of electrospun nanofibers composed of PVA/CS reinforced with amino-nanocellulose and loaded with DH, showing uniform morphology and nanoscale dimensions. (**B**) In vivo wound healing progression in rats treated with DH-loaded nanofiber membranes, demonstrating enhanced epithelial regeneration and reduced lesion size. Figures Reprinted from Arpita Priyadarshini Samanta, Adrija Ghosh, Koushik Dutta, Debashmita Mandal, Surajit Tudu, Kunal Sarkar, Beauty Das, Swapan Kumar Ghosh, Dipankar Chattopadhyay, Biofabrication of aminated nanocellulose reinforced polyvinyl alcohol/chitosan nanofibrous scaffold for sustained release of diltiazem hydrochloride, Pages No. 6, Copyright (2024), with permission from Elsevier [18].

**Figure 9 pharmaceutics-17-01491-f009:**
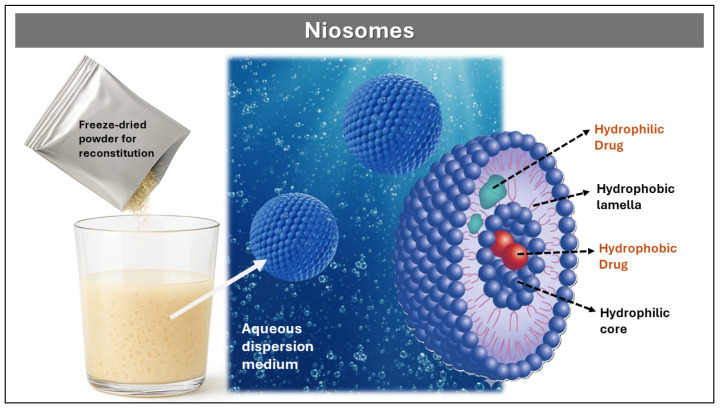
Schematic representation of niosome formation and structure. Freeze-dried powder is reconstituted in aqueous dispersion medium to form bilayer vesicles. Hydrophilic drugs are encapsulated in the aqueous core and surface, while hydrophobic drugs are embedded within the lipid bilayer.

**Figure 10 pharmaceutics-17-01491-f010:**
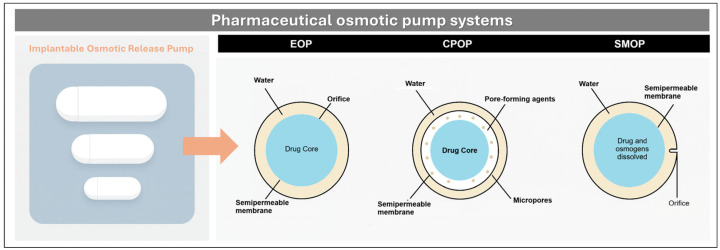
Schematic representation of pharmaceutical osmotic systems. (**Left**): implantable osmotic release pump. (**Right**): cross-sectional diagrams of EOP (semipermeable membrane with drilled orifice), CPOP (porogen-induced micropores), and SMOP (salt-modulated solubility gradient).

**Figure 11 pharmaceutics-17-01491-f011:**
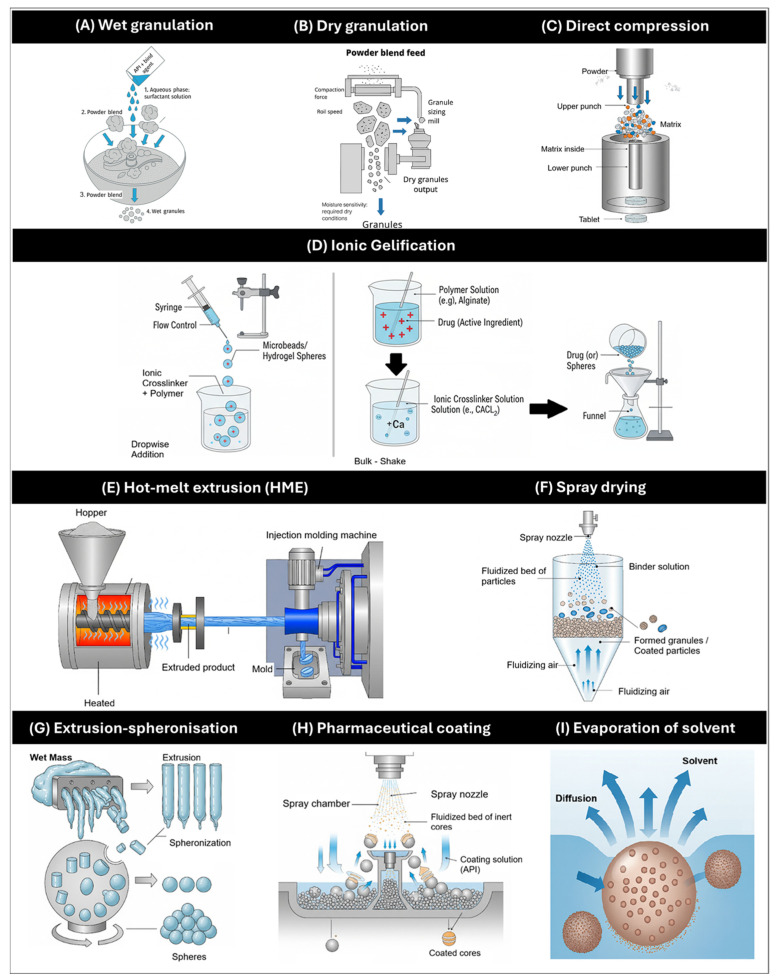
Schematic overview of advanced pharmaceutical technologies used in the design and optimization of modified-release and targeted drug delivery systems. The illustration includes: (**A**) Wet granulation; (**B**) Dry granulation; (**C**) direct compression; (**D**) ionic gelation; (**E**) Hot melt extrusion; (**F**) spray drying; (**G**) Extrusion-spheronization; (**H**) Pharmaceutical coating; (**I**) Evaporation solvent.

**Figure 12 pharmaceutics-17-01491-f012:**
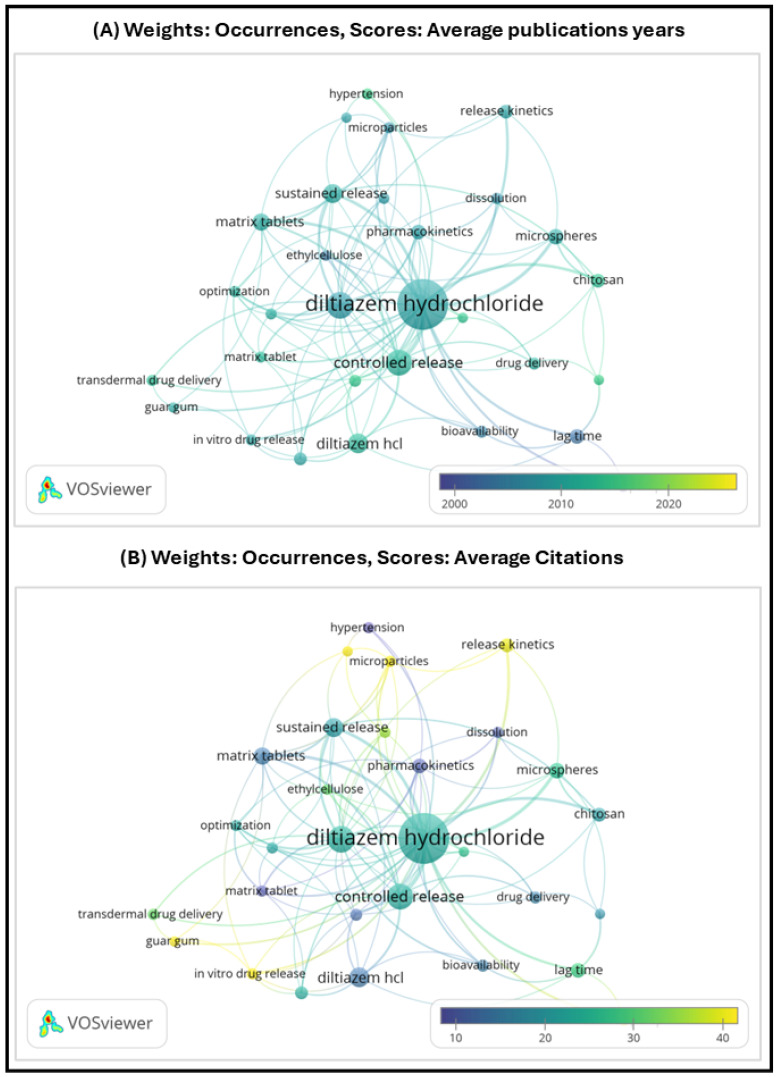
VOSviewer co-occurrence map generated from Scopus-indexed publications (2010–2025) using the keywords: Diltiazem hydrochloride, Emerging technologies, Methods of manufacture, and Modified release dosage forms. The clusters illustrate thematic concentrations in formulation strategies, manufacturing approaches, and innovation trends related to controlled release systems.

**Table 1 pharmaceutics-17-01491-t001:** Comparison between MDRFs and conventional dosage forms.

MDRFs	Advantages	Disadvantages	References
Matrix tablets	They can be produced in a wide variety of sizes and forms. Easy to manufacture. Increase the API’s stability to protect from hydrolysis into gastrointestinal tract.	Not favorable for APIs with poor solubility. Challenges in achieving uniform drug distribution and consistent release profile during the manufacture process. Gastric emptying, diet and other factors affect the release rate. Compatibility problems between API and polymeric material.	[60,61]
Nanosponges	Due to pore size (≤µm) bacteria cannot penetrate. They are stable at pH of 1–11 range and temperatures up to 300 °C. The scale-up process is easy; hence, they can be easily commercialized. The API is protected from the first-pass metabolism because of the use of crosslinkers.	The API loading capacity is altered by the crosslinking degree, which determinates the void space available. APIs must have a molecular weight between 100 and 400 Da and a melting point less than 250 °C.	[62,63]
Microsponges	They are free from harmful effects, non-irritating, non-mutagenic, and non-allergic. Adaptability to create innovative product shapes. They can prolong the API release up to 12 h.	API must not react with monomers and/or cause the preparation’s viscosity to rise while being formulated.	[64,65]
Pellets	They are less susceptible to dose dumping. Can be pellets with different release patterns in a single dosage form such as a capsule.	The use of granulating liquid such as water is necessary, requiring a drying phase and increasing the cost and time of manufacture.	[66]
Nanofibers	A high surface-to-volume ratio. Ease of fiber functionalization. Relatively low startup cost: A basic electrospinning system typically costs around $3000 to $4000.	The challenges in achieving in situ deposition of nanofibers on different substrates. It provides a low yield and needs a high working voltage. Little of material is deposited in terms of thickness, there is high electrical dispersion with high-conductive blends, and there are challenges with aqueous solutions and biomaterials.	[67]
Nanoparticles	Nanocarriers enhance solubility and bioavailability without altering the chemical structure of the API. Controlled and targeted release using stimuliresponsive nanocarriers.	Complex, expensive manufacturing with challenges in reproducibility and batch-to-batch consistency. Sensitive to environmental factors (temperature, pH, light), requiring specialized storage conditions. High research, development, and manufacturing costs due to advanced technologies.	[68,69]
Niosomes	Controlled shape, size, and composition. The API remains protected from gastrointestinal breakdown and first pass metabolism	Time-consuming process for niosome preparation. Physico-chemical instability.	[70,71]

**Table 2 pharmaceutics-17-01491-t002:** Comparative Analysis of Industrial Viability for MRDF Technologies.

Technology	Technological Level & Implementation	Industrial Viability	Regulatory Viability
Direct compression	Very basic/widely adopted	Very high (standard equipment, low cost)	Strong (QbD-ready, minimal validation)
Dry granulation	Basic/widely adopted	Very high (roller compaction, low cost)	Strong (simplified process, scalable)
Wet granulation	Basic/widely adopted	High (requires mixing and drying units)	Strong (CM-compatible, PAT integration)
Ionic gelification	Intermediate/selective use	Moderate (requires gelation setup)	Emerging (biopolymer-based, targeted systems)
Hot melt extrusion	Intermediate/growing adoption	High (specialized extruders, scalable)	Strong (solid dispersions, QbD, PAT)
Spray drying	Intermediate/widely used	High (atomization, scalable)	Strong (ASDs, inhalables, GMP-aligned)
Extrusion-spheronization	Intermediate/specialized	High (multiparticulates, dual equipment)	Strong (coating-ready, multiparticulate systems)
Pharmaceutical coating	Intermediate/standard practice	Very high (fluid-bed or pan coaters)	Strong (CR, enteric, masking, GMP compliant)
Solvent evaporation	Intermediate/common in R&D	Moderate (solvent recovery required)	Acceptable (depends on solvent and scale)
Combination methods	Advanced/case-dependent	Variable (requires integrated platforms)	Strong (flexible, QbD adaptable)
3D printing	Advanced/emerging	Low–moderate (limited infrastructure)	Limited (FDA-approved cases, evolving framework)
Osmotic systems	Advanced/specialized	Moderate–high (membrane design, modular)	Strong (IVIVC, lifecycle management)
IVIVC/PBPK modeling	Advanced/indirect	High (strategic design optimization)	Strong (VBE, biowaivers, post-approval support)

**Table 3 pharmaceutics-17-01491-t003:** Investigations from MRDF of DH.

Pharmaceutical Form	Preparation Method	Main Application	Composition Highlights	Reference
Buccal mucoadhesive tablets	Direct compression	Oral delivery; improve bioavailability	DH, mucoadhesive polymers (Carbopol-934, HPMC K4M, alginate, Na CMC, guar gum), talc, Mg stearate	[5]
PulsinCap^®^ system	Capsule crosslinking + wet granulation	Chronotherapy; latency-controlled release	DH granules, formaldehyde-crosslinked capsule, hydrogel plug (HPMC, ethylcellulose), disintegrants, MCC	[27]
Capsule tablets	Wet granulation + compression	Staged release; analgesic potential	DH, two tablets (fast/slow release), HPMC (50 and 4000 mPa·s), ethylcellulose, MCC, wheat starch	[9]
Sustained-release matrix tablet	Direct compression	24 h release; hypertension therapy	DH, Kollidon SR, HPMC K100, MCC, talc, Mg stearate	[26]
Nanoparticles	Ionic gelation	Enhance bioavailability; prolong half-life	DH, chitosan, Na TPP, Tween 80, glacial acetic acid	[115]
Pellets	Extrusion–spheronization	12 h sustained release	DH, MCC, CMC, demineralized water	[6]
Matrix pellets	Extrusion–spheronization	MCC-free sustained release for soluble drugs	DH, almond gum, Gelucire 43/01, lactose, water	[32]
Oroadhesive tablets	Direct compression	Prolong action; bypass hepatic metabolism	DH, kondagogu gum, guar gum, lactose, talc, Mg stearate	[116]
Electrospun nanofibers	Electrospinning	Transdermal delivery; wound healing	DH, PVA, CS, glutaraldehyde	[18]
Ionic liquid formulations	Solution preparation	Transdermal/topical vehicles	DH free base (from DH HCl + NaOH), ionic liquids	[117]
Microsponges	Evaporation diffusion	Rectal gels for anal fissures	DH, Eudragit RS100, DCM, PVA, methylcellulose or Poloxamer 407 hydrogels	[2]
Mucoadhesive microspheres	Spray drying	Nasal delivery; enhance residence and permeation	DH, low-MW chitosan, acetic acid, nitrous acid	[25]
Transdermal film/matrix	Solvent casting	Antihypertensive transdermal therapy	DH, HPMC K4M, Eudragit RS100, plasticizers (glycerol, DBP, PG), enhancers (cineole, capsaicin, DMSO, NMP)	[118]
Hydrophilic matrix	Direct compression	Stability and biopharmaceutical evaluation	DH, PEO (0.9–8 MDa), PSTPP, KCl, Na_2_CO_3_, talc, Mg stearate	[24]
Niosomes	Thin film hydration	Intranasal delivery; ↑ bioavailability and prolonged action	DH, Span 60, Brij-52, cholesterol	[57]
Drug-resin complex	Rotary bottle adsorption	Modified release using resin only	DH, Dowex 50WX8 resin, phosphate buffer	[23]
Matrix tablets	Wet granulation	SR profile comparable to commercial SR	DH, karaya gum, locust bean gum	[74]
Nanofibrous mats	Electrospinning	Wound healing; ↑ fibroblast proliferation, antioxidant effect	DH, PVA, glutaraldehyde, PBS	[19]
Hydrophilic matrix tablets	Direct compression	Avoid lag time; ~zero-order release	DH, PEO WSR 303, PEO N750	[119]
Nanoparticles	Water-in-oil emulsion	Pulmonary delivery; improved aerosolization	DH, CS, CS–L-leucine, glutaraldehyde, PBS	[34]
Nanofibrous mats	Electrospinning	Rapid/modulated release; hypertension	DH, RRP K30, HPMC K4M, aspartame, menthol, DMF	[120]
Matrix tablets	Wet granulation + compression	Oral SR; stable plasma levels	DH, PEC, guar gum, lactose, starch, Emcompress, Mg stearate	[75]
Microspheres	Emulsion–solvent evaporation	12–24 h release; reduced burst effect	DH, Eudragit RL/RS	[33]
Floating tablets	Direct compression	Gastroretention; prolonged gastric residence	DH, Methocel K4M/K15M, CS, Accurel, MCC, talc, Mg stearate	[121]
Tablets	Dry granulation + compression	SR tablets; in vitro evaluation	DH, HPC, HPMC, Eudragit/MAC	[76]
Nanoparticles	Ionotropic gelation	Oral SR; co-loaded with repaglinide	DH, CS, TPP, Tween 80, acetic acid	[115]
Microsponges	Solvent diffusion	Compatibility, amorphization, controlled release	DH (base/HCl), Eudragit RS100, organic solvents	[54]
Tablets	Compression + solvent coating	~8 h extended release; Ph. Eur. compliant	DH, NaCl, PEO/HPMC	[31]
Pull osmotic pump	Compression + coating	Solubility-independent delivery; IVIVC	DH, crosslinked capsule, PEO, NaCl, HPMC E5, PEG 4000, cellulose acetate	[22]
Mucoadhesive oral films	Pour molding	Oral retention; ↑ bioavailability	DH, PCSPh, aspartame, glycerol, propylene glycol	[122]
Matrix tablets	Direct compression	Robust SR under GI variability	DH, crosslinked potato starch (PI10), HPMC	[14]
Nanoparticles	Coating + solvent evaporation	Controlled release; reduced burst	DH, Silica-01/03, KH-570, acetone, capsules	[15]
Pellets	Extrusion–spheronization	MCC-free SR pellets	DH, Gelucire 43/01, almond gum	[32]
Microsponges	Quasi-emulsion solvent diffusion	Rectal gel systems	DH base, Eudragit RS100, PVA, DCM, ethanol	[10]
Tablets	Coating	Extended release; comparison with commercial tablets	DH, HPMC, CAP, acrylates, EC, PVP, NaCMC	[89]
Matrix tablets	Wet granulation	100% release in 12 h; ↑ bioavailability	DH, HEC, Na bicarbonate, lactose, PVP K30, talc, Mg stearate	[123]
Matrix tablets	Direct compression	24 h SR for hypertension, arrhythmia, angina	DH, HPMC, povidone, Tragacanth, Talc, Magnesium stearate, Lactose: fructose (1:1)	[124]
Floating matrix tablets	Direct compression	Extended release; ↑ gastric residence	DH, HPMC K4M/K15M, Na carbonate, lubricants	[125]
Pulsatile tablets	Capsular systems + pulsatile coatings	Chronotherapy (asthma, angina, hypertension)	DH, Eudragit, HPMC, polymeric alcohols, effervescent agents	[126]
Matrix tablets	Co-spray drying + compression	Enhanced SR vs. physical mixtures	DH, Kollicoat SR 30D, Kollidon SR, PVP	[127]
Matrix tablets	Wet granulation + compression	SR for hypertension/angina	DH, karaya gum, kondagogu gum, MCC, talc, Mg stearate	[128]
Matrix tablets	Wet granulation + compression	pH-independent SR	DH, casein, HPMC, lactose, purified water	[129]
Matrix tablets	Graft polymerization + granulation	Ca^2+^-responsive controlled release	DH, xanthan gum, acrylamide, ammonium persulfate, CaCl_2_	[130]
Bioadhesive buccal films	Solvent coating	Oral delivery; avoid first-pass metabolism	DH, PVA, PVP K30, Na CMC, glycerol	[131]
Electrospun nanocomposite membranes	Electrospinning	Wound healing; controlled release	DH, PVA, CS, PCL, ethanol, acetic acid, chloroform	[53]
Microspheres	Emulsion + crosslinking	pH-dependent controlled release	DH, sodium alginate, gelatin, glutaraldehyde, HCl	[49]
Nanosponges	Emulsion + solvent diffusion	Oral SR; ↑ bioavailability	DH, β-cyclodextrin, ethylcellulose, PVA, DCM, water	[8,47]
Floating pearls	External ionic gelation	Gastroretentive SR system	DH, Na alginate, CaCl_2_, sunflower oil, LMP	[8]
Osmotic tablets	Wet granulation + osmotic coating	Zero-order SR up to 24 h	DH, HPMC E3, lactose, MCC, cellulose acetate, PEG 400, triacetin, Mg stearate, silicon dioxide, Opadry^®^	[58]
Bilayer tablets	Direct layer compression	Bilayer gastroretentive SR system	DH, Avicel, lactose, Mg stearate, HPMC K4M, ethylcellulose, tragacanth, Na bicarbonate	[28]
Gelatinous microspheres	Ionic gelation	Sustained release; controlled in vitro profile	DH, gelatin, glutaraldehyde, water	[50]
Microspheres	Emulsion + compression	SR up to 12 h; reduced dissolution variability	DH, Eudragit RL100, RS100, RLPO, RSPO, Mg stearate	[11]

## Data Availability

No new data were created or analyzed in this study. Data sharing is not applicable.

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
