# Peer review of "Controlled Release Technologies for Diltiazem Hydrochloride: A Comprehensive Review of Solid Dosage Innovations"

_pharmaceutics, 2025, doi:10.3390/pharmaceutics17111491_

Round 1
Reviewer 1 Report
Comments and Suggestions for Authors
The present manuscript describes pharmaceutical technologies and formulation strategies applied to modified-release dosage forms (MRDF) of diltiazem hydrochloride. The manuscript is well structured and organized. It provides a comprehensive overview of aspects related to MRDF (release mechanism, patterns, and dosage forms), pharmaceutical formulations and technologies. The figures are clear and informative. I recommend the publication of the manuscript after a few modifications:
- I suggest including a more critical presentation of the examples of DH pharmaceutical formulations. Compare the examples and present advantages/disadvantages.
- Define abbreviations only the first time they are mentioned in the text.
- In figure 2B, it is a bit difficult to follow each pattern. I suggest changing the color or the line style for each pattern for better readability.
Reviewer 2 Report
Comments and Suggestions for Authors
The manuscript entitled "CONTROLLED RELEASE TECHNOLOGIES FOR DILTI-AZEM HYDROCHLORIDE: A COMPREHENSIVE REVIEW OF SOLID DOSAGE INNOVATIONS" by Troches-Mafla et al. covers an interesting arm of potential of sustained release or controlled release system in chronic conditions, such as hypertension. Diltiazem is a perfect candidate, with good physicochemical properties but poor half life. The authors have certainly covered works from 2010 to 2025 and explored opportunities within the delivery space, and emphasized on what are controlled release strategies employed or being researched for Diltiazem. In short how the traction is in this space. The authors however missed a fundamental aspect of literature review, discussing limitations and how the literature review could be used to show the real deal in science. In the current form I would recommend a rejection, since the review in current form is limited in utility. My specific comments and suggestions are as follows:
- While the authors want to discuss the aspect of controlled drug release, someone would expect a discussion on what is the ideal pattern for an ideal concentration in the plasma for an ideal response. And therefore, could be reproduced or reverse engineered. The review only discussed what strategies are being used and not exactly how close or far are these strategies to ideal drug release, and what can be done to make it ideal?
- The authors also miss a very important aspect, financial traction. How big of a market is for Diltiazem (We call it TAM) in US dollar, and what portion is covered by controlled release market? If it is really not investible for industry, because of multiple reasons, including lack of technology, market restrictions, slow growth etc. The advancement would only result in higher price per treatment for the patient, and the patient in a normal scenario would search for a cheaper option which is almost as efficacious.
- Why is reverse engineering this pattern so important? The answer to that should be, to show if really moving towards an ideal pattern necessary or viable? Given the therapeutic window, maybe an OROS tablet for which the technology exists in almost all major industrial manufacturing hubs, is similar in efficacy as compared to maybe nanoparticles? So, is it an unnecessary drag? Sometimes the answer is in safety vs efficacy! But the other aspect is real cost per tablet to a patient which is intertwined.
- The traction on one hand may be evident with discussion on financial aspects. The scientific traction may be visualized by that table before conclusion. But the best way to identify and visualize this traction would be with the help of a vos viewer diagram. The authors are recommended to add a vos viewer plot over the same time range and show how this technology is growing?
The authors are recommended to work on these suggestions. But the topic is niche, and the work has limited potential in my opinion and hence I would recommend a rejection in current form.
Reviewer 3 Report
Comments and Suggestions for Authors
The manuscript presents a comprehensive and well-structured review on Controlled Release Technologies for Diltiazem Hydrochloride. The content is scientifically sound, but the English language, phrasing, and stylistic consistency need moderate revision to ensure clarity, conciseness, and fluency suitable for publication in Pharmaceutics or similar journals.
Abstract
- L20–22: “...its short half-life and need for frequent dosing limit therapeutic adherence and increase plasma fluctuations.”
Suggest: “...its short half-life and frequent dosing requirements limit patient adherence and cause plasma concentration fluctuations.” - L23–25: “Objective: To critically review...”
Suggest merging: “This review critically examines recent pharmaceutical technologies and formulation strategies for modified-release dosage forms (MRDFs) of diltiazem hydrochloride, emphasizing their impact on pharmacokinetics, clinical performance, and regulatory aspects.” - L34–35: “Emerging platforms such as hot-melt extrusion with 3D printing and nanocarriers...”
Suggest: “Emerging platforms—such as hot-melt extrusion combined with 3D printing and nanocarriers—offer promising opportunities for personalized cardiovascular therapy.”
Introduction
- L49–50: “Additionally, DH exerts a negative chronotropic and inotropic effect by decreasing heart rate and prolonging diastolic duration...”
Suggest shortening: “DH also reduces heart rate and prolongs diastole, improving myocardial oxygen balance.” - L53–54: “Despite these favorable biopharmaceutical properties, conventional immediate-release formulations present significant clinical limitations.”
Suggest adding clarity: “Despite its favorable solubility and permeability, immediate-release DH formulations have notable clinical limitations.” - L70–73: “By integrating biopharmaceutical, technological, and clinical perspectives…”
Suggest: “This review integrates biopharmaceutical, technological, and clinical perspectives to provide a framework for developing safer and more effective DH formulations.”
Methodology
- L82–84: “The objective was to identify peer-reviewed studies describing manufacturing methods and formulation strategies…”
Suggest simplifying: “The objective was to identify peer-reviewed studies on manufacturing methods and formulation strategies for solid modified-release forms of DH.” - Consider adding a PRISMA-style flow description or number of studies screened/included for transparency.
Section 3: Modified Release Dosage Forms
- L98–100: “These MRDF can be classified based on three key criteria…”
“MRDFs can be classified according to (i) release mechanism, (ii) release pattern, and (iii) dosage form.” - L113–116: “At first, in diffusion-controlled systems...”
Replace “At first” with “In diffusion-controlled systems”. - L136–138: “From the perspective of release pattern…”
Suggest: “Based on their release patterns, MRDFs can be engineered to achieve specific pharmacokinetic profiles.” - L155–157: “Modified-release systems can be structurally classified into three main categories...”
Good sentence—no major issue. Consider adding a brief transition like “Structurally, MRDFs fall into three primary categories…”
Section 3.3.2 L217–219: “These units function as autonomous microreservoirs…”
Suggest splitting the long sentence for readability.
- L244–247: “Microsponges and nanosponges are porous particulate systems designed…”
“Microsponges and nanosponges are porous polymeric systems designed for controlled drug release.” - L299–304: “Microspheres are monolithic, dense particulate systems…”
Avoid repetition of “monolithic” (used earlier). Replace with “solid spherical particles composed of polymeric matrices.” - L396–404: “Nanofibers and nanoparticles are considered advanced platforms…”
“Nanofibers and nanoparticles represent advanced platforms for controlled release, offering high surface area and tunable physicochemical properties.” - L461–466: “Niosomes are multiparticulate vesicular systems formed by the self-assembly of non-ionic surfactants…”
Excellent description; however, avoid redundancy with later explanation. Merge the first two sentences.
Manufacturing Methods
- L603–612: “Wet granulation (Figure 11A) remains one of the most widely adopted techniques…”
Suggest cutting down adjectives: “Wet granulation remains widely used in DH MRDF development for enhancing compressibility and flow.” - L624–639: “Direct compression... streamlined and cost-efficient.”
Replace “Among the most streamlined” → “One of the most efficient and scalable techniques.” - L657–674: “Hot-melt extrusion (HME)...”
Suggest reducing redundancy: “HME enables continuous mixing and shaping of drug–polymer blends, allowing controlled release and integration with 3D printing for personalized dosage design.”
Section 4.10 (3D Printing) : Generally well written. Consider minor edits for conciseness:
- “Recent investigations have demonstrated the potential…”
“Recent studies demonstrate the potential of 3D printing to fabricate multi-drug dosage forms with programmable release.” - “This study underscores the potential…”
“This highlights the potential of HME-FDM integration for personalized MRDF fabrication.”
- Replace all instances of “DH-based formulations” with “diltiazem hydrochloride formulations” at first mention, then abbreviate thereafter.
- Ensure consistency in abbreviations (use MRDFs, not “MRDF” alone).
- Maintain uniform reference formatting (e.g., [21,22] or [21–23], but not mixed).
- Figures and tables: Verify that captions are self-contained and permissions are cited correctly.
The manuscript is generally well written and scientifically coherent. The technical terminology is accurate, and the authors demonstrate a strong command of subject-specific vocabulary. However, the overall quality of English requires moderate revision to improve clarity, conciseness, and readability. Several sentences are overly long or complex, and minor grammatical inconsistencies, redundant phrasing, and word-order issues occasionally hinder fluency. The authors are encouraged to simplify sentence structure, ensure consistent verb tense, and improve paragraph transitions for smoother flow. A careful language edit by a native or professional scientific editor is recommended to enhance clarity and polish the manuscript for publication.
Round 2
Reviewer 2 Report
Comments and Suggestions for Authors
The authors have managed to work extensively on the revisions and have improved the utility of this work a lot.
The authors are recommended to improve and revised the abstract and conclusion to suit the changes made. Also, I would recommend, the authors to amend a part of introduction to define the scope of review also covers reverse engineering ideal release pattern and integration of advanced techniques such as PBPK and IVIVC etc.
Author Response
Dear reviewer 2,
Thank you very much for your comments. We have improved the abstract, scope, and conclusions.
- Reviever comment: "The authors are recommended to improve and revised the abstract and conclusion to suit the changes made".
Author response: Suggestions were made.
2. Reviewer comment: " I would recommend, the authors to amend a part of introduction to define the scope of review also covers reverse engineering ideal release pattern and integration of advanced techniques such as PBPK and IVIVC etc".
Author response: Suggestions were made.
Sincerely
Yhors Ciro

Reviewer 3 Report
Comments and Suggestions for Authors
Author Response sufficicent
Author Response
Dear reviewer 3,
Thank you so much for the comments to improve our manuscript.
Sincerely
Yhors Ciro